# Genetic Variants Associated with Suspected Neonatal Hypoxic Ischaemic Encephalopathy: A Study in a South African Context

**DOI:** 10.3390/ijms26052075

**Published:** 2025-02-27

**Authors:** Caroline J. Foden, Kevin Durant, Juanita Mellet, Fourie Joubert, Jeanne van Rensburg, Khomotso Masemola, Sithembiso C. Velaphi, Firdose L. Nakwa, Alan R. Horn, Shakti Pillay, Gugu Kali, Melantha Coetzee, Daynia E. Ballot, Thumbiko Kalua, Carina Babbo, Michael S. Pepper

**Affiliations:** 1Institute for Cellular and Molecular Medicine, Faculty of Health Sciences, University of Pretoria, Pretoria 0084, South Africa; caroline.foden@gmail.com (C.J.F.); juanitamellet@yahoo.co.uk (J.M.); jeanne.vrensburg@gmail.com (J.v.R.); thumbiko.kalua@up.ac.za (T.K.); carina.babbo@up.ac.za (C.B.); 2BixBio Limited, Cape Town 8001, South Africa; kevin@bixbio.com; 3Centre for Bioinformatics and Computational Biology, Genomics Research Institute, Department of Biochemistry, Genetics, and Microbiology, University of Pretoria, Pretoria 0002, South Africa; fourie.joubert@up.ac.za; 4Department of Paediatrics and Child Health, Kalafong Hospital and Faculty of Health Sciences, University of Pretoria, Pretoria 0084, South Africa; khomotso.masemola@up.ac.za; 5Department of Paediatrics and Child Health, Chris Hani Baragwanath Academic Hospital, School of Clinical Medicine, Faculty of Health Sciences, University of the Witwatersrand, Johannesburg 2193, South Africa; sithembiso.velaphi@wits.ac.za (S.C.V.); firdose.nakwa@wits.ac.za (F.L.N.); 6Division of Neonatal Medicine, Department of Paediatrics and Child Health, Groote Schuur Hospital, University of Cape Town, Cape Town 7701, South Africa; alan.horn@uct.ac.za (A.R.H.); shakti.pillay@uct.ac.za (S.P.); 7Tygerberg Hospital Neonatal Unit, Department of Paediatrics and Child Health, Stellenbosch University, Cape Town 7600, South Africa; kali@sun.ac.za; 8Division of Neonatology, Department of Paediatrics and Child Health, Steve Biko Academic Hospital, Faculty of Health Sciences, University of Pretoria, Pretoria 0084, South Africa; mel.coetzee@up.ac.za; 9Department of Paediatrics and Child Health, Charlotte Maxeke Johannesburg Academic Hospital, Faculty of Health Sciences, University of the Witwatersrand, Johannesburg 2193, South Africa; daynia.ballot@wits.ac.za

**Keywords:** hypoxic ischaemic encephalopathy, genetic variants, whole genome

## Abstract

Neonatal encephalopathy suspected to be due to hypoxic ischaemic encephalopathy (NESHIE) carries the risk of death or severe disability (cognitive defects and cerebral palsy). Previous genetic studies on NESHIE have predominantly focused on exomes or targeted genes. The objective of this study was to identify genetic variants associated with moderate–severe NESHIE through whole-genome, unbiased analysis. Variant filtering and prioritization were performed, followed by association testing both on a case–control basis and to compare the grades of severity and/or progression. Association testing on neonates with NESHIE (N = 172) and ancestry-matched controls (N = 288) produced 71 significant genetic variants (false discovery rate corrected *p*-value < 6.2 × 10^−4^), all located in non-coding regions and not previously implicated in NESHIE. Disease-associated variants in non-coding regions are considered to affect regulatory functions, possibly by modifying gene expression, promoters, enhancers, or DNA structure. The most significant variant was at position 6:162010973 in the Parkin RBR E3 ubiquitin protein ligase (*PRKN*) intron. Intronic variants were also identified in genes involved in inflammatory processes (*SLCO3A1*), DNA repair (*ZGRF1*), synaptogenesis *(CNTN5)*, haematopoiesis *(ASXL2)*, and the transcriptional response to hypoxia *(PADI4)*. Ten variants were associated with a higher severity or lack of improvement in NESHIE, including one in *ADAMTS3*, which encodes a procollagen amino protease with a role in angiogenesis and lymphangiogenesis. This analysis represents one of the first efforts to analyze whole-genome data to investigate the genetic complexity of NESHIE in diverse ethnolinguistic groups of African origin and provides direction for further study.

## 1. Introduction

Neonatal encephalopathy (NE) is a clinical diagnosis made shortly after birth based on neurological signs, including an altered level of consciousness, abnormal reflexes, hypotonia, and often seizures [1]. NE is a complex condition with diverse causes, subtypes, and risk factors. Several conditions also mimic its clinical presentation [1,2]. Hypoxic ischaemic encephalopathy (HIE), a potential cause of NE, occurs due to an abnormal reduction in oxygenation and perfusion to the neonate’s brain during the acute peripartum period [3,4]. Determining the contribution of hypoxic ischemia to NE is challenging [2]. The cause of NE is often unknown, and a sentinel hypoxic event is infrequently present, despite fetal acidosis and/or low Apgar scores [1,5]. Until other causes of NE have been ruled out, the condition can be referred to as NE due to suspected HIE or NESHIE.

The severity of NESHIE can be graded according to clinical assessment systems such as the Thompson score [6] and modified Sarnat grading [7]. Neonates with moderate to severe NESHIE are at an increased risk of death or severe disability (cognitive defects, seizures, and cerebral palsy (CP)) [1,8]. In low- and middle-income countries, the incidence of NESHIE is significantly higher (1.5 to 20.3 per 1000 live births [9]) than the average in higher income countries of 1.5 per 1000 live births [10]. The incidence in South Africa is 2.3 to 13.3 cases per 1000 live births [8,11]. The standard of care for neonates with moderate or severe NESHIE and who are over 36 weeks gestational age is therapeutic hypothermia (TH). The neonate’s core body temperature is reduced to 33.5 for 72 h in order to provide neuroprotective benefits [4].

Phenotypic variability in CP has been ascribed to genetic factors [12], and such factors may similarly affect the incidence of NESHIE. In five whole-exome sequencing studies on a total of 1922 patients with CP, pathogenic or likely pathogenic genetic variants across 248 genes were identified [13,14,15,16,17]. However, classifying genetic variants as benign, pathogenic, or of uncertain significance [18] oversimplifies a complex issue, since the impact of other genetic, epigenetic, or environmental modifiers on such variants results in interactions that are unique to each individual and contribute to their risk profile [18,19].

Conditions that are clinically complex are frequently polygenic [20,21], and this may similarly be true for NESHIE. Complex diseases may result from disrupted and interconnected gene regulatory networks scattered across the genome [22]. The challenge lies in unravelling this complexity, compounded by the incomplete penetrance of associated genetic variants.

Translational medicine aims to identify genetic associations with complex diseases, to advance the goal of precision medicine in a clinical setting, and improve the quality of patient care [23,24]. Integrating patients’ genomic data with their clinical profiles is a powerful tool, facilitating improved advice in terms of prevention, prognosis, and educated reproductive decisions through screening [25]. It has been suggested that genomic sequencing be conducted in all children diagnosed with CP [26], neurodevelopmental disorders [27], and autism spectrum disorder [28]. Such an approach could direct treatment for patients with identifiable genetic contributors while enhancing the understanding of the genetic underpinnings of these complex conditions.

Historically, disease-associated genetic variants have been identified through the use of microarrays, which are limited to a small set of common, known variants [29], and exome studies, which exclude 99% of the genome [30]. However, the majority of variants associated with complex diseases are found in non-coding regions [20]. Expanding an investigation to include the entire genome therefore increases the potential to uncover novel associations. Whole-genome analysis, facilitated by genome-wide association studies (GWAS), has led to novel discoveries regarding the genetic basis of diseases including Crohn’s disease [31] and melanoma [32], amongst others [33]. However, GWAS require large sample sizes to achieve the required level of significance, given that individual variants typically explain only a small proportion of genetic variance. In addition, the multiple-testing burden is high [34]. An alternative approach is variant filtering and prioritization, where the number of variants tested is reduced to those considered more likely to have a functional effect by selecting those of low frequency in population databases and statistically increased in cases [35] when compared to ancestry-matched [36] controls.

This study reports on the results of an interim analysis of 172 neonates included in an ongoing investigation (the NESHIE study) into the molecular pathogenesis of NESHIE in neonates undergoing TH in South African public hospitals. The South African population is serviced by public and private healthcare sectors. The former serves more than 80% of the population and is less well resourced than the private sector. It is therefore critical that we focus on the sector that serves the majority of the population.

The primary objective of this study was to uncover genetic variants associated with NESHIE. Whole-genome studies provide an opportunity for unbiased analysis, and such studies on a large group of neonates diagnosed with NESHIE are lacking. We expect the genetics of NESHIE to be complex due to the clinically diverse nature of the condition. This research presents a crucial step toward unravelling the genetic underpinnings of NESHIE, leveraging the power of whole-genome analysis in an understudied population with high genetic diversity. Genomes of individuals from African ethnolinguistic groups contain more variation than those from other population groups [37] and consequently have high potential value in the search for genetic predispositions to complex diseases. These findings hold the promise of advancing our understanding of NESHIE, providing direction for further research, and contributing to the broader landscape of precision medicine.

## 2. Results

### 2.1. Data Set and Quality Control

Between June 2019 and September 2021, 702 neonates diagnosed with NESHIE were screened as per inclusion and exclusion criteria stated, with 268 enrolled into the study. Of these, a subset of neonatal samples (N = 174) was sent for whole-genome sequencing (WGS). Those selected for sequencing were the first neonates enrolled who had the required DNA quality following extraction. Two neonates were subsequently removed following DNA sequence quality control measures. In addition, 288 BixBio controls were included.

Table 1 lists the number of neonates in each clinical severity category. The baseline assessment represents the worst Sarnat grade and corresponding Thompson grouping prior to the initiation of TH. The Thompson score typically resulted in a milder severity categorization when compared to Sarnat grading, with some exceptions. The majority (60%) of neonates improved during the course of hospital admission with TH. Four neonates did not survive: three were graded as moderate Sarnat and Thompson at baseline, and one was graded as severe Sarnat and Thompson.

### 2.2. Variant Data Set

Mean genome sequencing coverage was relatively high: 31.02× for neonates and 28.21× for controls. Following variant calling and quality filtering, 45,360,793 single nucleotide variants (SNVs) and insertion/deletion (indel) variants were retained, averaging 5,155,185 per sample. Neonates and controls showed genetic overlap on a principal component analysis (PCA) plot (Figure 1A), indicating suitability for a case–control study. Ethnolinguistic diversity was observed among neonates (20) and controls (eight ethnolinguistic groups). These ethnolinguistic groups are represented in Figure 1B in a PCA plot, where the genetic diversity within these groups is apparent.

The full quality-filtered autosomal variant data set was employed for allele frequency (AF) filtering. After filtering variants with an AF < 0.02 in all controls and AF ≥ 0.05 in neonates, 4092 variants therefore considered semi-rare in controls and common in neonates with NESHIE were retained for association testing.

### 2.3. Association Testing—Case vs. Control

Association testing between neonates and BixBio controls was performed and a significance threshold of an FDR-corrected *p*-value of 6.2 × 10^−4^ (equivalent to a *p*-value of 1 × 10^−5^ in this data set) was selected following the visual inspection of the Manhattan plot (Figure 2) and guided by the calculation for the threshold for suggestive genome-wide significance proposed [38] of 1/(number of variants tested), equating to 2.44 × 10^−4^ in this data set. After confirmation and cross-checking, 71 significant variants remained. These are listed in Appendix A, together with their genomic location according to RefSeq 110 [39] annotation, AF, genotype counts, and the non-coding essential regulation (ncER) percentile. Odds ratios for the variants are shown in Appendix A. In summary, 39 variants were intergenic and 32 were located in introns. No significant variants were located in exons. The number of ethnolinguistic groups represented varied from four to twelve per variant, and therefore, none were considered to be ancestry specific.

The FDR-corrected *p*-values ranged from 6.18 × 10^−4^ to 8.40 × 10^−12^. Regarding variant type, eleven were deletions, two were insertions, and the balance were SNVs. The average AF was relatively low in the neonates (0.068 ± 0.016) and rare in the BixBio controls (0.0077 ± 0.0046). The number of significant variants found in each neonate ranged from zero (four neonates) to twenty-two (one neonate). Of the 71 variants, 6 were not present in the BixBio controls and 18 were not found in Gnomad v4. Within the BixBio controls, 150 individuals had none of these variants at all, while the maximum was 5 variants per control. Gene names for the variants located in introns, together with known or presumed gene functions, are listed in Table 2. The most significant variant, with an FDR-corrected *p*-value of 8.40 × 10^−12^, was an A-T SNV at position 6:162010973 in the Parkin RBR E3 ubiquitin protein ligase (*PRKN*) intron, one of four variants in this gene. Eight variants achieved an ncER percentile of higher than 80, implicating them in regulatory processes.

### 2.4. Allele Frequencies of Variants in Severity and Progression Groups

The 71 variants identified as significant on a case–control basis were tested across neonates grouped based on clinical severity before and after cooling as well as improvement or lack thereof. The allele frequencies for *p*-values < 0.05 are shown (Table 3). Among these variants, 10 were associated with a more severe and/or no improvement clinical course, whereas 12 were associated with a lower severity and/or improving clinical course. Two variants were associated with severe NESHIE on presentation but with improvement over time. Figure 3 indicates a subset of these results, namely the differences in allele frequencies at baseline for those variants that showed a significant association with either a moderate or severe grouping.

## 3. Discussion

This analysis represents one of the first efforts to analyze whole-genome data to investigate the genetic complexity of NESHIE in diverse ethnolinguistic groups of African origin. The analysis revealed 71 variants, including potential links to both severity and progression of NESHIE.

The NESHIE and CP genetics resource (NCGR) [41], established by our group, lists a number of genes and variants associated with HIE to date, reinforcing the genetically heterogeneous nature of this condition. There are no known monogenic causes of NESHIE. Rather, genetic factors may affect the neonate’s response to the trigger, namely hypoxia. None of the variants or genes identified in this study appear in the NCGR. This is likely due to the studies listed in the NCGR predominantly focusing on exons or a limited number of genes, or a lack of diversity (approximately 84% of the studies were in East Asian or European populations). This has resulted in a notable lack of studies into HIE that have searched for variants throughout the genome, or in diverse ancestries.

The variants identified in this study are located in intronic and intergenic regions, in line with evidence suggesting that a large proportion of disease-associated genetic variants are located in non-coding regions [20]. As our understanding of the human epigenome and transcriptome expands, it is likely that the impact of non-coding variants will become clearer. A 2017 study determined that 85% of investigated intergenic and intronic haploblocks containing GWAS SNPs are transcriptionally active [42].

Disease-associated variants in non-coding regions may impact regulatory functions [43]. Eight variants achieved a high ncER (non-coding essential regulation) percentile, implicating these variants in such processes. Variations in a single nucleotide may modify gene expression [44], promoters [45], or enhancers [46], or affect the state of chromatin and therefore DNA structure [47]. Long intergenic non-coding RNAs (lincRNAs), of which two were identified in this study, are associated with diagnosis, prognosis, and treatment response in malignancies, cardiometabolic disorders, and renal disease, among others [48]. Therefore, the regions of the genome that have historically been assumed to be non-functional and were excluded from studies can potentially hold great value. However, their functional interpretation is challenging.

While none of the variant-containing genes identified in this study have previously been implicated in HIE, several have functions that are of interest: *SLCO3A1* has been implicated in mediating inflammatory processes [49], *GMDS* is associated with cell adhesion and possibly neural development [50], *CNTN5* is involved in synaptogenesis [51], *ASXL2* is required for haematopoiesis [52], *DACH1* is involved in arteriogenesis [53], *PADI4* is required for the transcriptional response to hypoxia [54], and *ZGRF1* is involved in DNA repair [55].

Four *PRKN* variants identified in this study scored in the 91.97th percentile on ncER, indicating that they are implicated in the regulation of this gene. *PRKN* encodes Parkin, which is integral to the ubiquitin-mediated proteolysis pathway and is involved in the degradation of misfolded or damaged proteins and the reduction in inflammation, cell signalling, synaptic plasticity, and neuronal function [56]. Defects in Parkin are anticipated to be detrimental by exacerbating the impact of cellular stressors [57] such as hypoxia, where Parkin activation is crucial for mitophagy, essential for maintaining mitochondrial health. A previous study showed that neonates with severe HIE exhibit significantly elevated Parkin levels at birth, correlating with severity and further increasing over time [58].

In the absence of a sentinel event (for example, a ruptured uterus or placental abruption), identified risk factors for NESHIE are maternal (for example, maternal age over 35, diabetes, hypertension), antepartum (for example, maternal fever, chorioamnionitis), or intrapartum (for example, a prolonged second stage of labour) [59]. Environmental factors such as drug or alcohol use, malnutrition, poor living and working conditions, and communicable diseases may increase susceptibility to NESHIE by increasing stress in the developing fetal brain [60]. When combined with NESHIE risk factors or environmental fetal stressors, variants that affect the normal expression or regulation of the genes identified could conceivably lead to a poor response to hypoxia and subsequently the development of NESHIE.

Variants associated with severity and/or progression may indicate a different subtype of NESHIE when compared to those variants associated with a general NESHIE diagnosis. Eight variants were associated with severe NESHIE according to the Sarnat and/or Thompson group before cooling commenced, five intergenic variants and three located in introns, including a variant in *RBM6*, a gene involved in DNA repair [61]. The most significant variant is located in the Ferritin light chain pseudogene 10 (*FTLP10*) gene, one of four pseudogenes identified in this study. Long considered non-functional, increasing evidence indicates that pseudogenes can play vital regulatory roles [62]. The pseudogenes identified in this study have been identified as expressed in The Genotype-Tissue Expression project [63]. Pseudogene-derived lncRNAs have a high similarity to the transcripts of their parent genes, and therefore may act as competitors to sequester molecules, for example, miRNAs, and thereby affect homeostasis [64]. Transcriptomic profiling of both the pseudogene and its parent in neonates with and without the variant identified could elucidate a role in this condition.

The use of TH in the treatment of HIE has become a contentious issue in certain lower-to-middle income settings [65] following the publication of the results of the HELIX trial [66]. All neonates in this study underwent TH, and the majority (60%) improved from baseline to day 4/5, based on the grouped Thompson score. However, there were four genetic variants identified that were associated with the subgroup of neonates who were categorized as moderate or severe at day 4/5, and/or who did not improve during TH. These variants may therefore indicate a suboptimal response to TH. One of these variants, associated with both the more severe group and a lack of improvement, is located in the *ADAMTS3* intron. *ADAMTS3* encodes a procollagen amino protease that has been shown in a mouse model to play a role in angiogenesis and lymphangiogenesis [67]. The transcription of *ADAMTS3* is upregulated under hypoxic conditions in vitro [68]. Identifying variants associated with a lack of response to TH defines avenues for future research.

In conclusion, this study underscores the importance of unbiased whole-genome analysis in exploring the genetic underpinnings of NESHIE and contributes to both the understanding of the condition and to efforts toward enhancing precision medicine, particularly in underrepresented populations. The identified variants offer a foundation for further investigation into the genetic mechanisms contributing to this condition. Continued efforts to unravel the genetic complexity of NESHIE will enhance our ability to inform personalized therapeutic strategies and improve patient outcomes.

### Limitations and Future Work

While this study provides valuable insights, the AF-based filtering approach used may exclude some variants of importance. Although the sample size used is significant, particularly in a population that is historically severely underrepresented in genetic studies, it is relatively small when attempting to uncover genetic associations in complex diseases. As this is an interim analysis, results have not yet been validated in an independent cohort. The BixBio controls used were closely matched in ancestry and processed using the same pipeline in order to limit the chance of variant frequency differences being due to underlying genetic differences between cases and controls. However, some undetected biases may still remain.

The NESHIE study is ongoing. Future work will include repeating the analysis in a larger sample set and expanding the analyses to include a GWAS once sufficient sample numbers are obtained. The functional interpretation of non-coding variants is challenging and is generally accomplished through in silico analysis and/or functional studies, and by the addition of other data. The variants and genes identified in this study will be investigated further, with the inclusion of other omics data, including epigenetics, transcriptomics, and proteomics, and the further examination of the in-depth clinical information that is available for these neonates. In particular, a closer examination of the genetic and clinical picture of the neonates who did not improve following TH could provide insights into this at-risk subgroup.

## 4. Materials and Methods

### 4.1. Patient Recruitment

Participants were recruited from seven public hospitals in Gauteng and the Western Cape, South Africa. The seven public hospitals were Groote Schuur, Mowbray Maternity, Tygerberg, Steve Biko Academic, Charlotte Maxeke Johannesburg Academic, and Chris Hani Baragwanath Academic Hospitals. The study protocol followed, from screening to analysis, is summarized in Figure 4.

### 4.2. Inclusion and Exclusion Criteria

Neonates diagnosed with NESHIE were recruited following parental informed consent. Inclusion into this study was contingent on a gestational age of at least 36 weeks at the time of birth and a birth weight not less than 1800 g. Additionally, at least one biochemical or clinical feature indicative of hypoxic-related injury must be present. Biochemical markers for hypoxia included the following: a base deficit ≥ 16 and a pH ≤ 7, a base deficit ≥ 10, a pH ≤ 7.15, and/or evidence (within the first hour of life) of a peripartum or sentinel event. If early blood gas assessments were unavailable or not performed, clinical indicators suggestive of hypoxic injury included the following: a 5 min Apgar score of <7 or a need for resuscitation/assisted ventilation at 10 min of life. Additionally, at least one clinical sign suggestive of NE had to be present. Indicators for NE included either a lethargic, stuporous, or comatose state, a Thompson score of ≥7, or the presence of seizures (electrical and/or clinical). In the absence of seizures, one or more of the following presentations were required, in addition to a reduced level of consciousness, to indicate moderate–severe NE: decreased muscle tone (hypotonia/flaccid), abnormal/absent reflexes, and/or abnormal/absent suck.

If no exclusion criteria were present, participants were enrolled into the study following written informed consent. Exclusion criteria included the following: primarily non-hypoxic causes for the observed encephalopathy; neonates who did not undergo therapeutic hypothermia (TH), did not receive TH within 6 h of life, or were moribund and unlikely to benefit from TH; neonates who were asystolic or hypotensive or actively bleeding with no response to treatment; neonates with severe persistent pulmonary hypertension of the newborn (PPHN) with FiO_2_ > 80% not responding to standard treatment; neonates with congenital infections or severe abnormalities; and neonates suspected of having a severe chromosomal abnormality or surgical anomaly. Lastly, if consent was refused, not obtained prior to the demise of a neonate, or not obtained for any other reason, the participant was not enrolled into the study. Neonates had to be at least 24 h old before consent could be sought from parents to allow time for adequate counselling. Neonates born to mothers who were younger than 18 years were not included in this study.

### 4.3. Clinical Treatment and Monitoring

All included neonates were commenced on TH by 6 h of life using standard cooling methods, which included automated whole-body cooling using a cooling mat or a Servo-Controlled Gel-Pack to a core rectal temperature of 33.5 °C for 72 h, after which they were rewarmed by 0.2 to 0.5 °C every hour until their temperature reached 36 °C to 36.5 °C. Clinical data collected included but was not limited to amplitude integrated electroencephalography (aEEG), cranial ultrasound scan images, magnetic resonance imaging, and neurological examinations. Ethnolinguistic groups were identified through maternal race/first language.

Modified Sarnat (hereafter referred to as “Sarnat”) gradings and Thompson scores were collected and used to objectively assign a clinical grade of HIE at each time period, irrespective of aEEG. For early time points (baseline and pre-cool), Sarnat gradings and Thompson scores were determined. The retrospective estimation of Sarnat grades were determined from Thompson scores where this score was not reflected. Thompson scores were collected at later time points—daily for 7 days or until discharged.

For genetic association testing of severity and progression groups, the worst score assigned before cooling commenced was used to determine the “baseline” severity. Post-cooling severity was determined as the worst score obtained on day 4 or 5. Thompson scores were grouped into severity categories mild (1–6), moderate (7–14), or severe (≥15) [69]. Such Thompson categories have to date only been validated in the first 6 h. A neonate was assigned a category of “improved” if the Thompson group assigned post-cooling was of a lower severity than that assigned pre-cooling. If a neonate died before day 4 or 5 (2 neonates), a post-cooling grade of severe was assigned, and the neonate was categorized as not improved.

### 4.4. Ancestry-Matched Controls

There is substantial genetic variation within and between African ethnolinguistic groups, a fact that is becoming increasingly apparent and that has been unexposed initially by the shortage of studies including participants of African origin [37,70]. Therefore, the confirmation of sufficient genetic similarity between cases and controls is imperative in order to prevent confounding factors and bias due to ancestry. For this reason, ancestry-matched controls from BixBio (BixBio Limited; Cape Town, South Africa), consisting of South African individuals with no known major health concerns and hereafter referred to as BixBio controls, were utilized for association testing following approval from the BixBio Data Access Committee.

### 4.5. Blood Collection and DNA Isolation

Either venous umbilical cord (up to 3 mL) or peripheral blood (between 0.5 and 1 mL) was collected from neonates. Blood was collected into Zymo DNA/RNA Shield^®^ blood collection tubes (Zymo Research; Irvine, CA, USA) according to the manufacturer’s recommendations and stored at room temperature until informed consent was obtained (up to 14 days). After informed consent was obtained, the samples were shipped at ambient temperature to a professional biobank, Wits DIH (formerly Clinical Laboratory Services (CLS); Braamfontein, Johannesburg, South Africa). At the biobank, the samples were aliquoted and stored long term at −80 °C.

Aliquots were shipped to Inqaba Biotec (Inqaba Biotechnical Industries; Muckleneuk, Pretoria, South Africa) for isolation using the Quick-DNA^TM^ Miniprep Plus Kit (Zymo Research, Tustin, CA, USA) and an optimized version of the manufacturer’s protocol. DNA quality was confirmed. Isolated DNA elutes were stored at −20 °C and shipped on dry ice to an Illumina sequencing facility (Cambridge, UK) for sequencing.

### 4.6. DNA Sequencing and Variant Calling

Minor differences in DNA sequencing and variant calling pipelines may result in subtle but systemic differences in AF and subsequent false-positive associations [71,72]. Cases and controls were therefore processed using the same pipeline to allow for accurate comparisons.

Library preparation, sequencing, and variant calling were performed by BixBio, Cape Town, South Africa. Library preparation used the Illumina DNA PCR-free prep kit (Illumina, Cambridge, UK), while sequencing was performed using an Illumina NovaSeq 6000 platform. To prepare the reads for alignment, the adapter trimming of three or more matched bases was performed using Illumina DNA PCR-Free Prep adapter sequences. Additionally, quality trimming was performed with a minimum rolling average score of 12, and poly-G trimming was enabled to improve mapping reliability. The reads were then aligned to Illumina’s GRCh38-based multigenome graph reference, version 2. Alignment and variant calling were carried out using Illumina’s DRAGEN version 4.0. DRAGEN 4.0 achieved an F1 score of 0.9983 for Illumina sequencing data in the 2020 Precision FDA Truth Challenge V2, demonstrating superior accuracy among the pipelines and sequencing technologies tested.

### 4.7. Data Quality Control

Strengthening the Reporting of Genetic Association Studies (STREGA) [73] guidelines were followed, and SNP & Variation Suite v8.9.1 (Golden Helix, Inc., Bozeman, MT, USA) [74] was used for all quality control and analysis steps. Stringent quality control measures were implemented to ensure that only consistently high-quality variants and samples were retained.

To ensure the integrity of the data set, samples with a coverage of less than 10× following genome sequencing were excluded from further analysis due to the likelihood of missed variant calls. Subsequently, individual variant calls for single nucleotide variants (SNVs) and indels were filtered to remove those of poor call quality. For this process, DRAGEN’s machine-learning based variant score calibration was enabled. Variant calls that failed to meet the following criteria were removed: a quality score below 3; a genotype quality below 20; a read depth below 5; and/or an alternate read ratio (defined as the proportion of reads mapping to the alternate allele) falling outside the range of 0.3 to 0.7 for heterozygous calls or ≤0.85 for homozygous alternate calls. Following quality filtering, absent calls were assumed to represent homozygous reference alleles for the purposes of allele frequency calculations.

Samples were subsequently evaluated for outliers based on mean ± 3 standard deviations in the number of variant calls or heterozygosity rate. Samples exceeding these thresholds were likely to be of poor quality or indicative of sample contamination or sequencing errors and were therefore removed from the data set. This resulted in one neonate and 42 BixBio controls being removed. All samples left in the data set at this point were more than 15× coverage.

### 4.8. Relatedness

To prevent false-positive associations, samples that showed a high degree of genetic relatedness were identified and removed from the analysis. Linkage disequilibrium (LD) pruning was first performed, since relatedness calculations perform better if variants included are not in LD with each other. The composite haplotype method (CHM) was used for LD pruning with a window size of 50, a window increment of 5, and an LD threshold of r^2^ = 0.5. Subsequently, related individuals were identified using identity by descent analysis. This is a measure of how many alleles at a variant in each of two individuals came from the same ancestral chromosome. Sample pairs with a determined relatedness > 0.25, indicating 25% of alleles from the same ancestral chromosome and therefore second-degree relatedness, were identified. In each pair of related samples, the individual with the lower coverage was removed from further analysis. As a result of this, 12 samples, all from the BixBio control set, were removed.

### 4.9. Population Stratification

Guarding against population stratification is crucial in genetic studies, particularly when involving individuals of African descent, given the high genetic diversity within and between African ethnolinguistic groups. PCA was performed in order to identify potential genetic outliers and ensure sufficient genetic overlap between NESHIE and BixBio control groups to be considered suitable for AF testing. PCA was performed on the quality-filtered genomic data set after LD pruning, using a large data approach based on the Halko algorithm [75], with fixed random seed initialization, the additive genetic model, and normalized using theoretical standard deviation under the Hardy–Weinberg equilibrium. The first two principal components were plotted, and the resulting X-Y plot was visually inspected.

### 4.10. Variant Filtering and Prioritization

A variant filtering and prioritization approach was employed in this study. This narrows down the list of potential variants to be tested by selecting only those that are of high quality and at an AF more likely to be causative of the condition being studied [76]. This enables whole-genome analysis in limited sample sizes and allows the investigation of insertions and deletions, which are excluded from standard GWAS.

A variant found at an AF in the general population above that of the frequency of the condition being studied is considered not causative of that condition. However, this determination depends on the penetrance of the variant [76]. Since the penetrance of variants in this study is not known, a relaxed upper AF threshold of 0.02 in all controls was chosen. Variants located in autosomes with an AF < 0.02 in BixBio controls and Gnomad v4 [77], and ≥0.05 in NESHIE neonates, were therefore retained to prepare a variant set more likely to consist of relevant variants.

### 4.11. Variant Association Testing

Association testing (alternate vs. reference allele) of NESHIE neonates vs. BixBio controls was performed in order to identify significant differences in alternate vs. reference allele frequencies between neonates and controls, without assuming any underlying genetic model. Fisher’s Exact Test was used as recommended for small sample sizes, with the false discovery rate (FDR) for multiple-testing correction. Ancestry distribution per variant was inspected to identify potential ancestry biases. The supervised machine learning model ncER genome-wide percentile score [78] of each variant was determined to identify those implicated in regulatory processes. The higher the percentile, the more likely that the variant is essential in regulation. Association testing of the set of significant variants was subsequently conducted for severity categories both before and after TH, as well as for improvement or lack thereof of the condition.

## Figures and Tables

**Figure 1 ijms-26-02075-f001:**
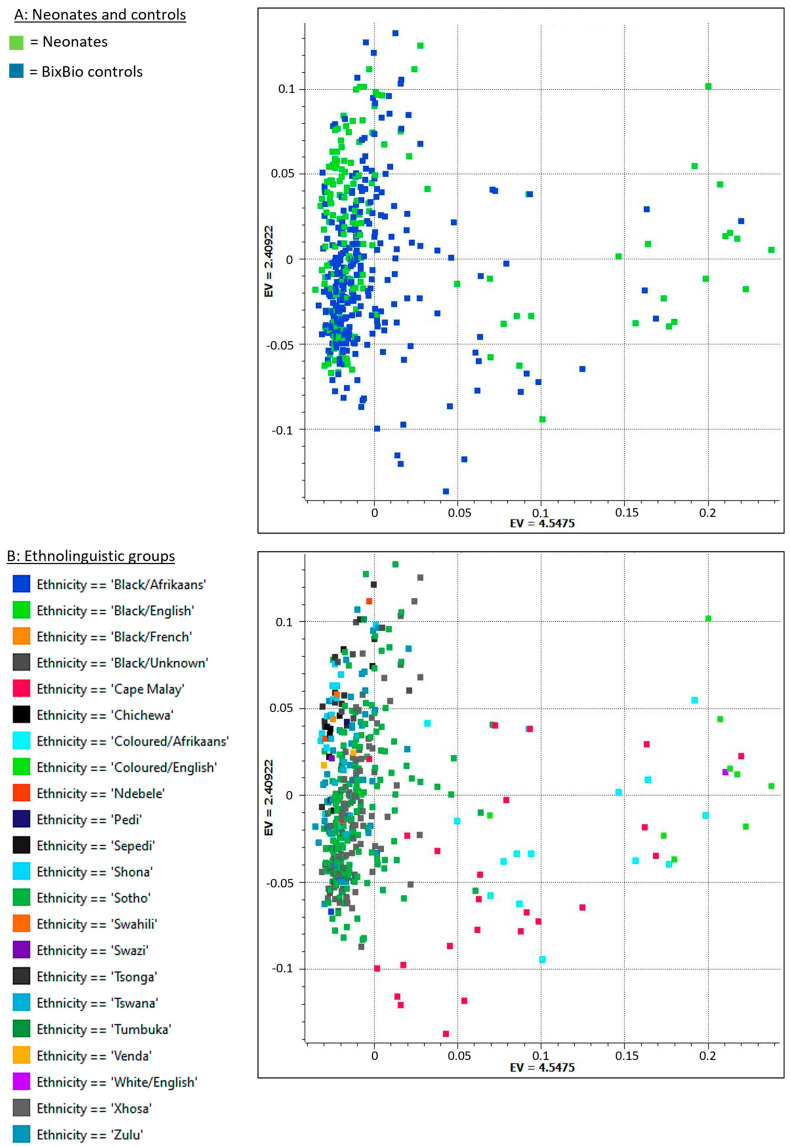
Genetic diversity of study participants: Principal component analysis plot of the first two principal component eigenvalues (EVs). (**A**) represents neonates and controls, and (**B**) represents ethnolinguistic groups.

**Figure 2 ijms-26-02075-f002:**
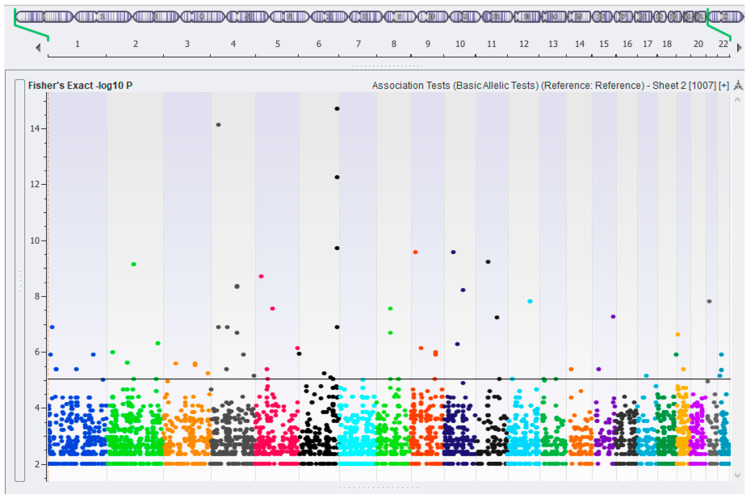
Manhattan plot illustrating filtered variants of interest, coloured per chromosome: The Y axis represents the *p*-value (−log10), and the X axis indicates the position of the variants in the genome, chromosome 1 (**left**) to 22 (**right**). Each dot represents a variant. Variants exceeding the chosen significance threshold (FDR-corrected *p* < 6.2 × 10^−4^) are indicated above the horizontal line.

**Figure 3 ijms-26-02075-f003:**
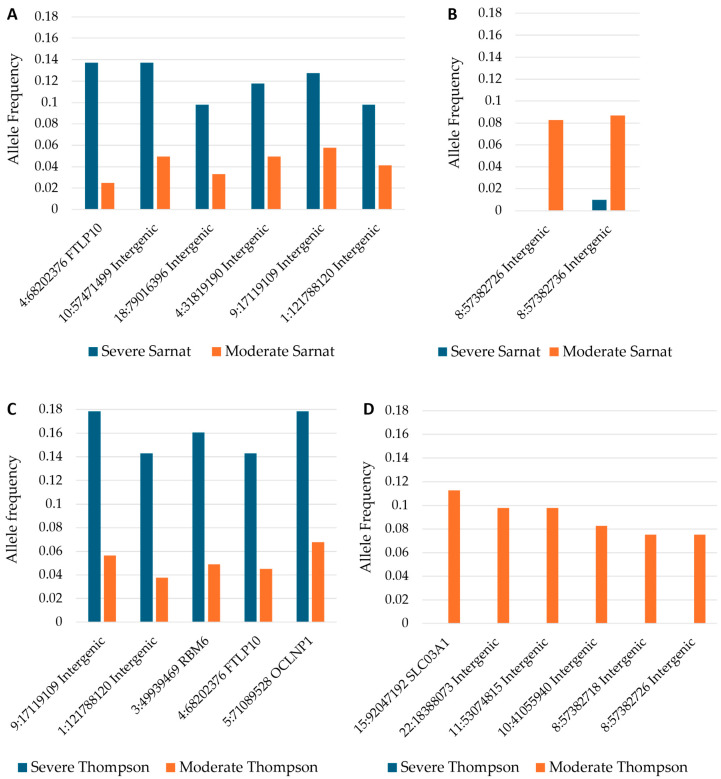
Variants significantly associated with severity categories: Variants with an allele frequency significantly (*p* < 0.05) higher (**A**,**C**) or lower (**B**,**D**) in neonates with severe NESHIE at baseline when compared to neonates with moderate NESHIE. Severity determination based on the Sarnat grade is shown in (**A**,**B**), while that based on the Thompson score is shown in (**C**,**D**). Short variant names (chromosome:position) and location or gene are indicated.

**Figure 4 ijms-26-02075-f004:**
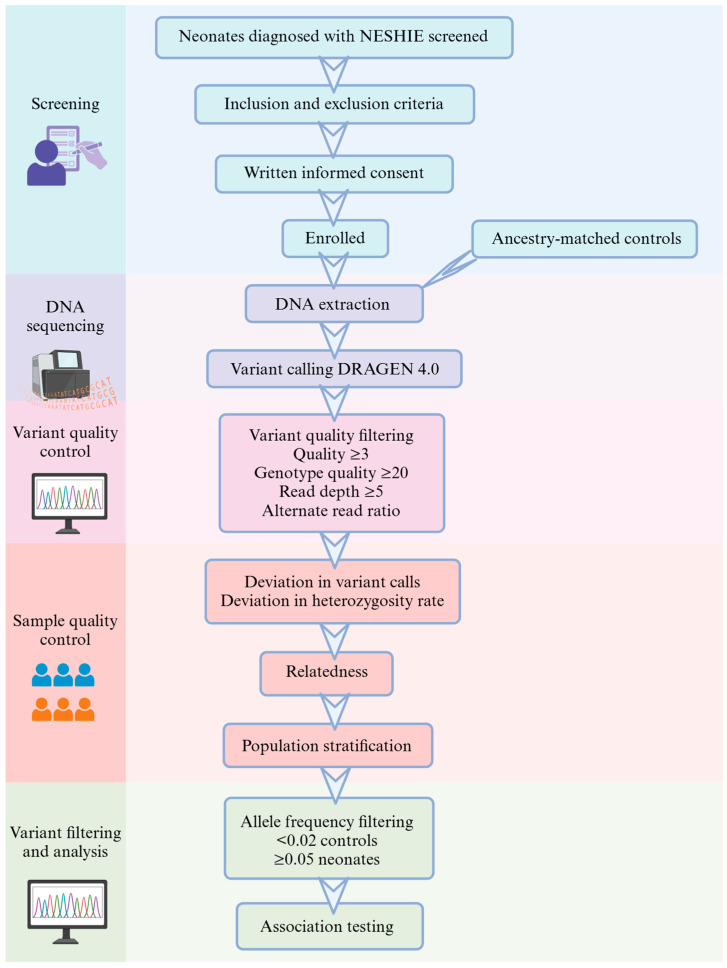
Summary of sample selection and analysis protocol.

**Table 1 ijms-26-02075-t001:** Number of neonates displaying the respective severity and progression categorizations at baseline and various time points.

Severity Categories of Neonates (N = 172)	Number (%) Normal	Number (%) Mild	Number (%) Moderate	Number (%) Severe	Number (%) Not Measured ^a^
Sarnat baseline	0 (0)	0 (0)	121 (70)	51 (30)	0 (0)
Thompson baseline	0 (0)	11 (6)	133 (77)	28 (16)	0 (0)
Thompson day 4/5 ^b^	7 (4)	81 (47)	71 (41)	10 (6)	3 (2)
**Progression Categories of** **Neonates (N = 172)**	**Number (%) Improved ^c^**		**Number (%) Not Improved**		**Number (%) Not Measured ^a^**
Thompson progression baseline to day 4/5 ^b^	104 (60)		65 (38)		3 (2)

^a^ “Not measured” designation indicates that the Thompson score was not determined at that time point. ^b^ Thompson group for the later time point was determined as worst grade on day 4 or 5. ^c^ Neonates were categorized as improved if the worst grouped Thompson score on day 4 or 5 was of lower severity than that at baseline.

**Table 2 ijms-26-02075-t002:** Variants located in introns, including those of protein-coding genes, pseudogenes, and non-coding RNA genes.

Number of Variants in the Gene	Gene Symbol	Gene Name	Known or Suspected Gene/Gene Product Function ^a^
1	*ADAMTS3*	ADAM metallopeptidase with thrombospondin type 1 motif 3	Protease, a role in the processing of type II fibrillar collagen in articular cartilage
1	*ANAPC1P4*	ANAPC1 pseudogene 4	Pseudogene
1	*ASXL2*	ASXL transcriptional regulator 2	Epigenetic regulator, binds histone-modifying enzymes and involved in the assembly of transcription factors
1	*CDC73*	Cell division cycle 73	RNA polymerase II core binding
1	*CNTN5*	Contactin 5	Glycosylphosphatidylinositol (GPI)-anchored neuronal membrane protein that functions as a cell adhesion molecule
2	*CPNE4*	Copine 4	Calcium-dependent, phospholipid-binding protein, may be involved in membrane trafficking, mitogenesis, and development
1	*DACH1*	Dachshund family transcription factor 1	Chromatin-associated protein, associates with other DNA-binding transcription factors to regulate gene expression and cell fate determination during development
1	*FGD6*	FYVE, RhoGEF and PH domain containing 6	Guanyl-nucleotide exchange factor activity, small GTPase binding
1	*FTLP10*	Ferritin light chain pseudogene 10	Pseudogene
1	*GMDS*	GDP-mannose 4,6-dehydratase	Conversion of GDP-mannose to GDP-4-keto-6-deoxymannose, using NADP+ as a cofactor
1	*INVS*	Inversin	Interacts with nephrocystin and infers a connection between primary cilia function and left–right axis determination
1	*LINC02679*	Long intergenic non-protein coding RNA 2679	lncRNA
2	*LOC102724710*	Uncharacterized LOC102724710	ncRNA
1	*NAT16*	N-acetyltransferase 16 (putative)	Predicted to enable acyltransferase activity, transferring groups other than amino-acyl groups
1	*OCLNP1*	Occludin pseudogene	Pseudogene
1	*PADI4*	Peptidyl arginine deiminase 4	Enzyme responsible for the conversion of arginine residues to citrulline residues
1	*PDSS2*	Decaprenyl diphosphate synthase subunit 2	Enzyme that synthesizes the prenyl side chain of coenzyme Q, one of the key elements in the respiratory chain
4	*PRKN*	Parkin RBR E3 ubiquitin protein ligase	A component of a multiprotein E3 ubiquitin ligase complex that mediates the targeting of substrate proteins for proteasomal degradation
1	*RBM6*	Rap associating with DIL domain	Enables GTPase binding activity
1	*SLCO3A1*	Solute carrier organic anion transporter family member 3A1	Sodium-independent organic anion transmembrane transporter activity
2	*THRAP3*	Thyroid hormone receptor-associated protein 3	Enables phosphoprotein binding, thyroid hormone receptor binding, and transcription coactivator activity
1	*TJP3*	Tight junction protein 3	Role in linkage between the actin cytoskeleton and tight junctions, also sequesters cyclin D1 at tight junctions during mitosis
1	*ULK4P3*	ULK4 pseudogene 3	Pseudogene
3	*ZGRF1*	Zinc finger GRF-type containing 1	GRF zinc fingers are found in a number of DNA-binding proteins

^a^ As per the National Centre for Biotechnology Information Gene database [40].

**Table 3 ijms-26-02075-t003:** Genetic variants associated with severity and/or progression of NESHIE.

Variant	Gene Region (Gene If Applicable)	Allele Frequencies for Severity, Severe (n = 51) vs. Moderate (n = 121) Sarnat at Baseline, *p* < 0.05	Allele Frequencies for Severity, Severe (n = 28) vs. Moderate (133) Thompson at Baseline, *p* < 0.05. Missing n = 11 ^a^	Allele Frequencies for Severity, Severe or Moderate (n = 81) vs. Mild or Normal (n = 88) Thompson Day 4 or 5, *p* < 0.05. Missing n = 2 ^b,c^	Did Not Improve (n = 65) vs. Improved (n = 104) Thompson from Baseline to Worst Grade Day 4 or 5. Missing n = 3 ^b^	Associated with More Severe and/or Not Improving NESHIE	Associated with Milder and/or Improving NESHIE	Associated with More Severe NESHIE at Baseline but Improvement over Time
NC_000001.11:g.121788120C>T	Intergenic	0.098 vs. 0.041 (*p* = 0.047)	0.143 vs. 0.038 (*p* = 0.006)			x		
NC_000002.12:g.114110393C>T	Intergenic			0.086 vs. 0.028 (*p* = 0.031)	0.092 vs. 0.034 (*p* = 0.029)	x		
NC_000004.11:g.31819190G>A	Intergenic	0.118 vs. 0.050 (*p* = 0.035)				x		
NC_000004.12:g.68202376A>C	Intron (*FTLP10*)	0.137 vs. 0.025 (*p* = 1.44 × 10^−4^)	0.143 vs. 0.045 (*p* = 0.012)			x		
NC_000004.12:g.72456033C>T	Intron (*ADAMTS3*)			0.099 vs. 0.034 (*p* = 0.025)	0.108 vs. 0.039 (*p* = 0.021)	x		
NC_000005.10:g.71089528A>G	Intron (*OCLNP1*)		0.179 vs. 0.068 (*p* = 0.016)			x		
NC_000010.11:g.57471499C>T	Intergenic	0.137 vs. 0.050 (*p* = 0.007)				x		
NC_000010.11:g.79642025T>A	Intron (*LINC02679*)			0.148 vs. 0.057 (*p* = 0.006)		x		
NC_000018.10:g.79016396C>T	Downstream intergenic	0.098 vs. 0.033 (*p* = 0.030)				x		
NC_000019.10:g.30075028_30075029del	Intergenic				0.108 vs. 0.029 (*p* = 0.004)	x		
NC_000002.12:g.87718514G>A	Intron (*ANAPC1P4*)			0.025 vs. 0.085 (*p* = 0.018)	0.015 vs. 0.082 (*p* = 0.013)		x	
NC_000006.12:g.162011042_162011043del	Intron (*PRKN*)				0.031 vs. 0.096 (*p* = 0.028)		x	
NC_000008.11:g.57382718G>A	Intergenic		0 vs. 0.075 (*p* = 0.031)	0.025 vs. 0.085 (*p* = 0.018)			x	
NC_000008.11:g.57382726del	Intergenic	0 vs. 0.083 (*p* = 0.002)	0 vs. 0.075 (*p* = 0.031)	0.025 vs. 0.085 (*p* = 0.018)			x	
NC_000008.11:g.57382736T>A	Intergenic	0.010 vs. 0.087 (*p* = 0.007)					x	
NC_000010.11:g.41055940C>T	Intergenic		0 vs. 0.083 (*p* = 0.019)	0.025 vs. 0.102 (*p* = 0.004)	0.015 vs. 0.096 (*p* = 0.003)		x	
NC_000011.10:g.53074815C>T	Intergenic		0 vs. 0.098 (*p* = 0.012)				x	
NC_000012.12:g.95148880_95148887del	Intron (*FGD6*)				0.015 vs. 0.077 (*p* = 0.013)		x	
NC_000013.11:g.71735277T>C	Intron (*DACH1*)			0.025 vs. 0.085 (*p* = 0.018)			x	
NC_000015.10:g.92047192C>T	Intron (*SLCO3A1*)		0 vs. 0.113 (*p* = 0.004)	0.062 vs. 0.136 (*p* = 0.029)	0.046 vs. 0.135 (*p* = 0.009)		x	
NC_000022.11:g.18388073G>A	Upstream intergenic		0 vs. 0.098 (*p* = 0.012)				x	
NC_000022.11:g.18732630G>A	Intergenic			0.025 vs. 0.080 (*p* = 0.029)	0.015 vs. 0.077 (*p* = 0.013)		x	
NC_000003.12:g.49939469C>T	Intron (*RBM6*)		0.161 vs. 0.049 (*p* = 0.006)		0.023 vs. 0.091 (*p* = 0.013)			x
NC_000009.12:g.17119108_17119116del	Intergenic	0.128 vs. 0.058 (*p* = 0.046)	0.179 vs. 0.056 (*p* = 0.005)		0.015 vs. 0.120 (*p* = 3 × 10^−4^)			x

^a^ A missing value for baseline Thompson scores was due to the neonate being scored as mild (association testing here was moderate vs. severe). ^b^ A missing value for later severity and progression testing was due to the required severity categorization not being determined for those neonates at that time point. ^c^ Since most neonates improved clinically during the first 4–5 days, severity testing for the later time point was conducted on moderate or severe vs. mild or normal.

## Data Availability

Genetic variant calls in .csv format, per individual for both neonates and BixBio controls, are available for the allele frequency-filtered data set, i.e., the data set used to perform association testing and produce the results in this study. These data are available on reasonable request from the corresponding author and provided the proposed use of the data is approved by the Research Ethics Committee of the University of Pretoria Medical School. The sharing of full whole-genome data sequences has not been approved by the Research Ethics Committee, due to the personal nature of genomic data and the vulnerable nature of the study population, i.e., neonates with a severe health condition for whom consent was provided by their parent(s). The data offered to be shared are sufficient to reproduce the results of this study or conduct further NESHIE-related analysis.

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
