# Peer review of "Genetic Variants Associated with Suspected Neonatal Hypoxic Ischaemic Encephalopathy: A Study in a South African Context"

_ijms, 2025, doi:10.3390/ijms26052075_

Round 1

Reviewer 1 Report

Comments and Suggestions for Authors

The current manuscript presents innovative data on the genetic variants associated with NESHIE, a clinically complex condition, which genetic underpinning  fits a polygenic inheritance with several affected loci involved in  interconnected regulation of gene networks. WGS  of 172 affected neonates and 288 ancestry-matched controls and filtering of the variants by association testing (case vs control) and according to grades of disease  progression led the Authors to  identify 71 significant genetic variants, all novel (not present in the NCGR resource the Authors had previously established).  Intriguingly all disease-associated variants map to non coding regions of the genome (intronic and intergenic regions) implying they may impact gene regulation, promoters or enhancers or chromatin state. These results are enticing and highlight a pipeline for further studies.

I have a comment regarding the title of  Table 2   “ Variants located in introns”   considering that  pseudogenes (formed from both protein coding genes and non coding genes)  often lack introns and  2 long intergenic non coding RNA  are also included in the table. However according to Suppl Table 2 my view is not updated.

As regards the sentence “ Eight variants achieved a high ncER percentile  : add into brackets “non-coding essential regulation”  for reader non familial with GWAS

Figure 3: the 12 variants  significantly associated with severity categories are indicated by their chromosome position which is technically perfect, though it is  cryptic  for the reader to recognize the involved gene, if any.  it would be  helpful to discuss  other variants associated with NESHIE  severity besides PRNK, FTLP10 and ADAMTS3.

Author Response

Comment 1: The current manuscript presents innovative data on the genetic variants associated with NESHIE, a clinically complex condition, which genetic underpinning fits a polygenic inheritance with several affected loci involved in interconnected regulation of gene networks. WGS of 172 affected neonates and 288 ancestry-matched controls and filtering of the variants by association testing (case vs control) and according to grades of disease progression led the Authors to identify 71 significant genetic variants, all novel (not present in the NCGR resource the Authors had previously established).  Intriguingly all disease-associated variants map to non-coding regions of the genome (intronic and intergenic regions) implying they may impact gene regulation, promoters or enhancers or chromatin state. These results are enticing and highlight a pipeline for further studies.

Response: Thank you for contributing your time and expertise towards reviewing our manuscript. We appreciate your insight and are pleased you found the results enticing.

Comment 2: I have a comment regarding the title of Table 2 “Variants located in introns” considering that pseudogenes (formed from both protein coding genes and non-coding genes) often lack introns and 2 long intergenic non-coding RNA are also included in the table. However, according to Suppl Table 2 my view is not updated.

Response: Thank you for your comment. Please note that Table 2 is now labelled as Table 3. According to RefSeq annotation and the information available for the pseudogenes and ncRNAs included in this study, they all do possess intron and exon (although non-coding) structures – with the introns presumably being spliced out following transcription to create a mature transcription product. Therefore, this terminology was used to indicate the position of the variant within the gene, according to RefSeq classification. To clarify this point, we have changed the title of Table 3 (page 7) to be “Variants located in introns, including those of protein-coding genes, pseudogenes, and non-coding RNA genes”.

Comment 3: As regards the sentence “Eight variants achieved a high ncER percentile: add into brackets “non-coding essential regulation” for reader non familial with GWAS

Response: Thank you for pointing this out. This has been added as requested (discussion paragraph 4, page 12).

Comment 4: Figure 3: the 12 variants significantly associated with severity categories are indicated by their chromosome position which is technically perfect, though it is cryptic for the reader to recognize the involved gene, if any.  

Response: Agreed. Figure 3 has been updated (page 11) as suggested to reflect chromosome position as well as location (intergenic or the gene name where relevant).

Comment 5: It would be helpful to discuss other variants associated with NESHIE severity besides PRNK, FTLP10 and ADAMTS3.

Response: Thank you for the suggestion. The discussion has been expanded as suggested (discussion, paragraph 8, page 12) to discuss all variants associated with higher severity.

Reviewer 2 Report

Comments and Suggestions for Authors

Review for the manuscript ,,Genetic variants associated with suspected neonatal hypoxic ischemic encephalopathy: A study in a South African context,,

The manuscript described neonatal encephalopathy due to due to hypoxic ischemic encephalopathy (NESHI),one of the most frequent neonatal diseases in South Africa. The authors tried to identify genetic variants associated with NESHIE using whole genome sequence data. They also analysed these variants according to the severity and/or progression of the diseases. They identified 71 suspected genetic variants, 10 in intronic regions of the genes.

The manuscript provide contribution for the management of neonates with NESHIE.

It is a well done study, with much details on the methods used in order to be replicated in others laboratories.

Given the severity of the disease and its high incidence, the results help clinicians to manage the factors that could improve the quality of life in this medical condition, to manage severe disability.

I also note a very good presentation of the results and conclusions of this study.  

Author Response

Comment 1: The manuscript described neonatal encephalopathy due to hypoxic ischemic encephalopathy (NESHI), one of the most frequent neonatal diseases in South Africa. The authors tried to identify genetic variants associated with NESHIE using whole genome sequence data. They also analysed these variants according to the severity and/or progression of the diseases. They identified 71 suspected genetic variants, 10 in intronic regions of the genes. The manuscript provide contribution for the management of neonates with NESHIE. It is a well done study, with much details on the methods used in order to be replicated in others laboratories. Given the severity of the disease and its high incidence, the results help clinicians to manage the factors that could improve the quality of life in this medical condition, to manage severe disability. I also note a very good presentation of the results and conclusions of this study.  

Response: Thank you for contributing your time and expertise towards reviewing our manuscript. We appreciate your comments and feedback and are pleased you found the study valuable and well done.

Reviewer 3 Report

Comments and Suggestions for Authors

The study of Foden at all. Sheds a new light on the Neonatal encephalopathy suspected to be due to hypoxic-ischemic encephalopathy (NESHIE) - in contrast to other studies it focuses on the genome, not on the selected genes. The patients have been recruited in 7 public southafrican hospitals. Association testing on neonates with NESHIE (N=172) and ancestry-matched controls (N=288) produced 71 significant genetic variants (false discovery rate corrected p-value < 6.2 x 10-4), all located in intronic and intergenic regions The variants were linked to the severity and progression Moreover, several intronic variants have been identified and discussed. The introduction gives background on the complexity and describes it as a clinical condition. In the disussion complexity of the disease, demografic heterogenicity and the impact of non-codinf variants are mentioned.

Introduction:

  1. Are there any monogenic causes of NESHIE?
  2. Has any of the child been diagnosed with any otehr genetic disorder?
  3. Is there any heretiablity of the condition?
  4. I am not fully convinced of the significance of non-coding variants. Have they been assessed in silico? how is it possible to narrow down their number?
  5. Abbreviation TH should be spelled out and described shortly
  6. Please describe the role of environmental factors in the condition and discuss in the context of the genetic findings.

Author Response

Comment 1: The study of Foden at all. sheds a new light on the neonatal encephalopathy suspected to be due to hypoxic-ischemic encephalopathy (NESHIE) - in contrast to other studies it focuses on the genome, not on the selected genes. The patients have been recruited in 7 public South African hospitals. Association testing on neonates with NESHIE (N=172) and ancestry-matched controls (N=288) produced 71 significant genetic variants (false discovery rate corrected p-value < 6.2 x 10-4), all located in intronic and intergenic regions. The variants were linked to the severity and progression. Moreover, several intronic variants have been identified and discussed. The introduction gives background on the complexity and describes it as a clinical condition. In the discussion, complexity of the disease, demographic heterogenicity and the impact of non-coding variants are mentioned.

Response: Thank you for contributing your time and expertise towards reviewing our manuscript.

Comment 2: Are there any monogenic causes of NESHIE?

Response: Thank you for your question. In our review of the literature, no cases of NESHIE that have been attributed to monogenic disorders were found. This has been clarified in the discussion, paragraph 2 (page 11).

Comment 3: Has any of the child been diagnosed with any other genetic disorder?

Response: Thank you for the question. In our study, no apparent genetic disorders were noticed clinically at birth and during follow-up. However, some genetic disorders may be less apparent and/or manifest later in life.

Comment 4: Is there any heritability of the condition?

Response: Thank you for the question. Heritability has not been assessed in our study population to date. It has been suspected but not directly assessed in any other NESHIE study that we know of. It is likely that genetic variants affect the response to an environmental trigger, namely hypoxia, which is (thankfully) not present in all infants in the peripartum period. It would, therefore, be difficult to accurately assess heritability since multiple infants and/or multiple generations would have to be affected and exposed to the trigger (hypoxia) in a given family.

Comment 5: I am not fully convinced of the significance of non-coding variants. Have they been assessed in silico? How is it possible to narrow down their number?

Response: Thank you for your question. We have expanded the relevant discussion points as follows. We agree the functional interpretation of non-coding variants is challenging, as stated in the introduction and reinforced in section 4, paragraph 2 (page 13). The studies referenced that discuss the potential significance of non-coding variants (paragraph 4 of the discussion, page 12) reached their conclusions through detailed in silico analysis, functional studies (for example mouse models), or a combination of both. The number of variants of interest can be narrowed down through replication in an independent cohort, expanding the study in a larger sample size, and/or by integrating other ‘omics (for example transcriptomics) and detailed clinical data. The use of transcriptomic profiling is discussed in paragraph 8 of the discussion (page 13) and future work planned to investigate the identified variants in more detail is discussed in section 4 (page 13).  

Comment 6: Abbreviation TH should be spelled out and described shortly.

Response: Thank you for the suggestion. A definition and brief description have been added to the second paragraph of the introduction (page 3).

Comment 7: Please describe the role of environmental factors in the condition and discuss in the context of the genetic finding.

Response: Thank you for the suggestion. We agree that environmental factors are important and have added a paragraph discussing risk factors for NESHIE, including environmental factors, and discussed this in the context of our findings (discussion, paragraph 7, page 12). While research into environmental risk factors for NESHIE is lacking when compared to pregnancy or labour complications such as placental issues or preeclampsia, the importance of environmental factors is likely to become increasingly clear, particularly in a resource-poor setting such as South Africa. Future work will include assessing the variants identified in the context of clinical data and environmental factors, where such data is available. For example, we have recently undertaken whole genome bisulfite sequencing of DNA from NESHIE neonates, which we hope will shed light on the environmental/epigenetic contribution.

Reviewer 4 Report

Comments and Suggestions for Authors

Overall this manuscript presents a novel and important contribution to our understanding of the genetic underpinnings of “NESHIE” in a diverse African population. The use of whole genome sequencing and the identification of potentially relevant non-coding variants are significant strengths. However, the manuscript would benefit from a more structured discussion, clearer emphasis on clinical implications, and a more comprehensive exploration of limitations and future directions. Addressing these points will enhance the impact and relevance of this valuable work.

 Abstract

- The authors should consider restructuring the abstract to delineate background, methods, results, and conclusions more clearly.

- The objective could be more precisely stated. Instead of "to identify genetic variants associated with NESHIE throughout the genome," it could specify whether the aim was to identify novel variants, confirm known ones, or both.

- While the abstract mentions 71 significant variants, it doesn't provide any specific examples. Including one or two of the most significant findings could enhance impact.

- The authors should provide a clear statement on the significance of these findings. How do these results advance the field or potentially impact clinical practice?

- The conclusion that "variants identified offer a foundation for further investigation" is somewhat weak. A stronger statement about the potential implications of these findings would be more appropriate.

 Introduction

-  The term NESHIE (Neonatal Encephalopathy Suspected to be due to Hypoxic Ischemic Encephalopathy) could be more clearly defined early in the introduction.

- The authors could strengthen their justification for using whole genome sequencing rather than exome sequencing or targeted approaches. While they mention that disease-associated variants are often found in non-coding regions, they could elaborate on why this is particularly relevant for NESHIE.

- Given the study's focus on "South African public hospitals", the introduction could provide more context on the specific challenges or unique aspects of studying NESHIE in this population.

- The introduction could more explicitly state what gap in the current knowledge this study aims to fill, highlighting its novelty in the field of NESHIE genetics.

- While the objective is stated, the introduction could benefit from a clear hypothesis about what types of genetic variants the authors expect to find associated with NESHIE.

- The authors are recommended to briefly explain why their interim analysis is important and what it aims to contribute at this stage.

- This section could more clearly articulate the potential clinical or research implications of identifying genetic variants associated with NESHIE.

Methods

- The authors could include a brief explanation of how they determined the sample size for this interim analysis (172 neonates).

- More details on how the BixBio controls were selected and their characteristics (e.g., age, sex distribution) could be provided in this context.

- The authors could provide more details on the statistical methods used for the association testing, including the software used.

- The authors should consider including a power analysis to demonstrate that the sample size is sufficient to detect meaningful associations.

- More information on how variants were annotated (e.g., which databases or tools were used) could be helpful.

- The authors should provide more details on the specific settings and parameters used in the bioinformatics tools to enhance reproducibility.

-  If there were any missing data points, the authors should explain how these were handled in their analysis.

- The authors should consider if any validation methods (e.g., replication in an independent cohort) were used or planned for significant findings.

Results

- To facilitate following the authors in their elaboration, please consider adding a flow diagram to illustrate the sample selection process, from screening to final analysis.

- For the association testing results, consider reporting effect sizes (e.g., odds ratios) in addition to p-values.

-  The rationale for choosing the specific FDR-corrected p-value threshold (6.2 x 10-4) was not clear in the text.

- From the referee's point of view, the supplementary tables include informative data that should be presented as main tables; in particular, this journal has no restriction for the number of included Figures/Tables to support this section. Also, it is enough to consider three decimals for each value presented in these tables to avoid number crowdness.

Discussion

This section is comprehensive and provides a thoughtful interpretation of the study's findings. However, there are several areas where improvements can be made to enhance clarity and depth.

- It is highly recommended to start with a stronger statement summarizing the novelty and significance of the study (e.g., "This study represents one of the first efforts to apply whole genome sequencing to investigate NESHIE in a diverse African population, uncovering novel insights into its genetic underpinnings.")

- The authors speculate on the regulatory roles of non-coding variants but do not discuss whether any functional validation (e.g., gene expression studies) is planned or feasible. Adding this would strengthen the implications of their findings.

- The authors should expand on how their findings could impact clinical practice or therapeutic strategies for NESHIE. For example, how might these variants inform personalized medicine or treatment decisions like therapeutic hypothermia?

- The discussion on TH is interesting but somewhat brief. Could these genetic findings help identify neonates who are less likely to respond to TH? This could be an important clinical implication worth expanding on.

- The mention of pseudogenes (e.g., FTLP10) is intriguing but underexplored. The authors could elaborate on why pseudogenes might play a role in NESHIE and how they plan to investigate this further.

- It is recommended to discuss how integrating other omics data (e.g., transcriptomics, epigenomics) might provide additional insights into NESHIE pathogenesis.

- It is recommended to add a sentence or two discussing how non-coding variants might be prioritized for functional studies (e.g., CRISPR-based approaches or transcriptomic analyses).

- By the end of this section, the authors should conclude with a stronger emphasis on how this research contributes to global efforts in precision medicine, particularly for underrepresented populations.

The limitation section

This section  could be expanded to discuss other potential issues, such as:

  • The inability to validate findings in an independent cohort.
  • The challenge of interpreting non-coding variants without functional data.
  • Potential biases introduced by ancestry-matched controls from BixBio.

Author Response

Comment 1: Overall this manuscript presents a novel and important contribution to our understanding of the genetic underpinnings of “NESHIE” in a diverse African population. The use of whole genome sequencing and the identification of potentially relevant non-coding variants are significant strengths. However, the manuscript would benefit from a more structured discussion, clearer emphasis on clinical implications, and a more comprehensive exploration of limitations and future directions. Addressing these points will enhance the impact and relevance of this valuable work.

Response: Thank you for contributing your time and expertise towards reviewing our manuscript. We are pleased you find the work valuable and appreciate your detailed comments.

Comment 2: Abstract - The authors should consider restructuring the abstract to delineate background, methods, results, and conclusions more clearly. The objective could be more precisely stated. Instead of "to identify genetic variants associated with NESHIE throughout the genome," it could specify whether the aim was to identify novel variants, confirm known ones, or both. While the abstract mentions 71 significant variants, it doesn't provide any specific examples. Including one or two of the most significant findings could enhance impact. The authors should provide a clear statement on the significance of these findings. How do these results advance the field or potentially impact clinical practice? The conclusion that "variants identified offer a foundation for further investigation" is somewhat weak. A stronger statement about the potential implications of these findings would be more appropriate.

Response: Thank you for your suggestions. We agree, and the abstract has been modified as suggested.

Comment 3: Introduction - The term NESHIE (Neonatal Encephalopathy Suspected to be due to Hypoxic Ischemic Encephalopathy) could be more clearly defined early in the introduction. The authors could strengthen their justification for using whole genome sequencing rather than exome sequencing or targeted approaches. While they mention that disease-associated variants are often found in non-coding regions, they could elaborate on why this is particularly relevant for NESHIE. Given the study's focus on "South African public hospitals", the introduction could provide more context on the specific challenges or unique aspects of studying NESHIE in this population. The introduction could more explicitly state what gap in the current knowledge this study aims to fill, highlighting its novelty in the field of NESHIE genetics. While the objective is stated, the introduction could benefit from a clear hypothesis about what types of genetic variants the authors expect to find associated with NESHIE. The authors are recommended to briefly explain why their interim analysis is important and what it aims to contribute at this stage. This section could more clearly articulate the potential clinical or research implications of identifying genetic variants associated with NESHIE.

Response: Thank you. The introduction has been modified as suggested and changes have been marked. The term NESHIE requires that the introduction and background for its use in this study be described before it is defined, and therefore the description and definition have been left unchanged (paragraph 1 of the introduction).

Comment 3: Methods.

Comment 3a: The authors could include a brief explanation of how they determined the sample size for this interim analysis (172 neonates).

Response: Thank you for your comment. This is discussed in section 2.1 of the results (page 4) – the samples included were the first set recruited that had the required DNA quality for the study.

Comment 3b: More details on how the BixBio controls were selected and their characteristics (e.g., age, sex distribution) could be provided in this context.

Response: Thank you for the suggestion. Given that this study looked at autosome genetic allele frequencies, the age and sex of the controls were not relevant. Considering the high levels of genetic diversity among individuals of African ethnolinguistic groups, the most important consideration when selecting appropriate controls was to determine sufficient genetic similarity to the neonates, in order to reduce bias. In addition, cases and controls must be processed using the same pipeline. These points have been emphasised in section 5.4 relating to ancestry-matched controls (page 15) and further discussed in sections 5.6 (DNA sequencing page 16, first paragraph) and 5.9 (population stratification page 17).

Comment 3c: The authors could provide more details on the statistical methods used for the association testing, including the software used.

Response: Thank you for the comment. The software used is discussed in section 5.6 (Golden Helix SNP & Variation Suite). This software is, however, not required to reproduce the results as the methods (quality control, variant filtering, and association testing) are standard practice and could be performed using many tools and pipelines. More details on the association testing have been added to section 5.11 (page 17).

Comment 3d: The authors should consider including a power analysis to demonstrate that the sample size is sufficient to detect meaningful associations.

Response: Thank you. A power analysis is not possible since the effect sizes (delta) cannot be known or estimated at this stage, particularly in a complex condition. As an example, a delta of 0.4 produces a very high power of 0.95 in a sample size of 163, with a significance level of 0.05. However, more work is required before the true effect sizes of the variants identified may be discovered. The methodology chosen in this study, which reduces the number of variants to test based on allele frequency filtering, and the careful matching of cases and controls, increase the power regardless of effect size.

Comments 3d and 3e: More information on how variants were annotated (e.g., which databases or tools were used) could be helpful. The authors should provide more details on the specific settings and parameters used in the bioinformatics tools to enhance reproducibility.

Response: Thank you. Variants were annotated using RefSeq (results section 2.3, page 6). Details on parameters used for analysis are included throughout (for example variant quality thresholds in section 5.7 (page 16) and parameters used for LD pruning in section 5.8 (page 16)). We believe the methodology and parameters shared are suitable for reproducibility.

Comment 3f: If there were any missing data points, the authors should explain how these were handled in their analysis.

Response: Thank you, we agree. These are included in the manuscript. Missing clinical data: Sarnat grades were determined based on Thompson scores where the Sarnat score was not recorded (section 5.3, paragraph 2, page 15). Missing variant calls were assumed to represent homozygous reference calls (section 5.7, paragraph 2, page 16). This errs on the side of caution and avoids over-representing alternate allele frequencies. No other missing data points were present.

Comment 3g: The authors should consider if any validation methods (e.g., replication in an independent cohort) were used or planned for significant findings.

Response: Thank you for the suggestion. We agree, and this has been discussed in the “limitations and next steps” section (section 4, paragraph 1, page 13).

Comment 4: Results

Comment 4a: To facilitate following the authors in their elaboration, please consider adding a flow diagram to illustrate the sample selection process, from screening to final analysis.

Response: Thank you for the suggestion. We agree, and this has been added to the beginning of the materials and methods section (page 14).

Comment 4b: For the association testing results, consider reporting effect sizes (e.g., odds ratios) in addition to p-values.

Response: Thank you for the suggestion. These have been added as a supplementary file (Supplementary Table 1) and referred to in section 2.3 (first paragraph, page 6).

Comment 4c: The rationale for choosing the specific FDR-corrected p-value threshold (6.2 x 10-4) was not clear in the text.

Response: Thank you, we agree. This has been clarified as suggested in section 2.3 (paragraph 1, page 6). The threshold was chosen following visual inspection of the Manhattan plot of the variants and taking into consideration guidelines proposed in previous studies.

Comment 4d: From the referee's point of view, the supplementary tables include informative data that should be presented as main tables; in particular, this journal has no restriction for the number of included Figures/Tables to support this section. Also, it is enough to consider three decimals for each value presented in these tables to avoid number crowdness.

Response: Thank you, we agree. The decimals have been adjusted as suggested. The first table (Table 2) contains too many columns to fit legibly into the manuscript itself for review purposes, but we have included it as a separate file and referred to it as Table 2 (bottom of page 6). The second table has been inserted in the manuscript (Table 4, page 9).

Comment 5: Discussion

Comment 5a: It is highly recommended to start with a stronger statement summarizing the novelty and significance of the study (e.g., "This study represents one of the first efforts to apply whole genome sequencing to investigate NESHIE in a diverse African population, uncovering novel insights into its genetic underpinnings.")

Response: Thank you. We agree, and this has been corrected as suggested (first paragraph of the discussion, page 11).

Comment 5b: The authors speculate on the regulatory roles of non-coding variants but do not discuss whether any functional validation (e.g., gene expression studies) is planned or feasible. Adding this would strengthen the implications of their findings.

Response: Thank you for your comment. We agree and have added a discussion of planned future work to section 4 (page 13).

Comment 5c,d,e: The authors should expand on how their findings could impact clinical practice or therapeutic strategies for NESHIE. For example, how might these variants inform personalized medicine or treatment decisions like therapeutic hypothermia? The discussion on TH is interesting but somewhat brief. Could these genetic findings help identify neonates who are less likely to respond to TH? This could be an important clinical implication worth expanding on. The mention of pseudogenes (e.g., FTLP10) is intriguing but underexplored. The authors could elaborate on why pseudogenes might play a role in NESHIE and how they plan to investigate this further.

Response: Thank you, we agree. The point on TH is emphasised in the discussion (paragraph 9, page 13) and in limitations and future work (section 4, page 13). The possible role of pseudogenes has been expanded in the discussion, paragraph 8, page 12/13).

Comment 5f,g: It is recommended to discuss how integrating other omics data (e.g., transcriptomics, epigenomics) might provide additional insights into NESHIE pathogenesis. It is recommended to add a sentence or two discussing how non-coding variants might be prioritized for functional studies (e.g., CRISPR-based approaches or transcriptomic analyses).

Response: Thank you. We agree, and this has been added to the discussion (paragraph 8, page 13) and future work (section 4, page 13) sections.

Comment 5h: By the end of this section, the authors should conclude with a stronger emphasis on how this research contributes to global efforts in precision medicine, particularly for underrepresented populations.

Response: Thank you. This has been added as suggested, at the end of the discussion.

Comment 6: Limitations and future work: This section could be expanded to discuss other potential issues, such as:

  • The inability to validate findings in an independent cohort.
  • The challenge of interpreting non-coding variants without functional data.
  • Potential biases introduced by ancestry-matched controls from BixBio.

Response: Thank you for the suggestions. We have added as suggested to this section (page 13).

Reviewer 5 Report

Comments and Suggestions for Authors

Comments on the ijms-3476811 manuscript where a genomic study and variants in genes associated with NESHIE were identified. The background contains the necessary information on the subject of study, the objective of the study is determined, the material and methods are described in detail, the statistical analysis is applied appropriately, the results are adequately described in the tables and figures presented, and the discussion and conclusions are in agreement with the results obtained. The study contains novel information on the subject of NESHIE and the information can be applied to other neurological diseases. In general, the study seems complete and well described.

Author Response

Comments on the ijms-3476811 manuscript where a genomic study and variants in genes associated with NESHIE were identified. The background contains the necessary information on the subject of study, the objective of the study is determined, the material and methods are described in detail, the statistical analysis is applied appropriately, the results are adequately described in the tables and figures presented, and the discussion and conclusions are in agreement with the results obtained. The study contains novel information on the subject of NESHIE and the information can be applied to other neurological diseases. In general, the study seems complete and well described.

Response: Thank you for contributing your time and expertise towards reviewing our manuscript. We appreciate your comments.

Round 2

Reviewer 3 Report

Comments and Suggestions for Authors

Thank you for adressing the comments. I am satisified with answers. 

Reviewer 4 Report

Comments and Suggestions for Authors

Thanks to the authors for addressing the raised concerns.